# Empirical and Computational Evaluation of Hemolysis in a Microfluidic Extracorporeal Membrane Oxygenator Prototype

**DOI:** 10.3390/mi15060790

**Published:** 2024-06-15

**Authors:** Nayeem Imtiaz, Matthew D. Poskus, William A. Stoddard, Thomas R. Gaborski, Steven W. Day

**Affiliations:** 1Rochester Institute of Technology, Kate Gleason College of Engineering, Rochester, NY 14623, USA; ni2631@rit.edu (N.I.); wsbme@rit.edu (W.A.S.); trgbme@rit.edu (T.R.G.); 2Department of Bioengineering, University of Pittsburgh, Pittsburgh, PA 15261, USA; mdp77@pitt.edu

**Keywords:** microfluidics, hemolysis, computational modeling, multi-physics modeling, CFD, UDF

## Abstract

Microfluidic devices promise to overcome the limitations of conventional hemodialysis and oxygenation technologies by incorporating novel membranes with ultra-high permeability into portable devices with low blood volume. However, the characteristically small dimensions of these devices contribute to both non-physiologic shear that could damage blood components and laminar flow that inhibits transport. While many studies have been performed to empirically and computationally study hemolysis in medical devices, such as valves and blood pumps, little is known about blood damage in microfluidic devices. In this study, four variants of a representative microfluidic membrane-based oxygenator and two controls (positive and negative) are introduced, and computational models are used to predict hemolysis. The simulations were performed in ANSYS Fluent for nine shear stress-based parameter sets for the power law hemolysis model. We found that three of the nine tested parameters overpredict (5 to 10×) hemolysis compared to empirical experiments. However, three parameter sets demonstrated higher predictive accuracy for hemolysis values in devices characterized by low shear conditions, while another three parameter sets exhibited better performance for devices operating under higher shear conditions. Empirical testing of the devices in a recirculating loop revealed levels of hemolysis significantly lower (<2 ppm) than the hemolysis ranges observed in conventional oxygenators (>10 ppm). Evaluating the model’s ability to predict hemolysis across diverse shearing conditions, both through empirical experiments and computational validation, will provide valuable insights for future micro ECMO device development by directly relating geometric and shear stress with hemolysis levels. We propose that, with an informed selection of hemolysis parameters based on the shear ranges of the test device, computational modeling can complement empirical testing in the development of novel high-flow blood-contacting microfluidic devices, allowing for a more efficient iterative design process. Furthermore, the low device-induced hemolysis measured in our study at physiologically relevant flow rates is promising for the future development of microfluidic oxygenators and dialyzers.

## 1. Introduction

The prevalence of chronic lung diseases, such as Chronic Obstructive Pulmonary Disease (COPD), and unpredictable and potentially overwhelming outbreaks of acute infectious illnesses, such as swine flu and COVID-19, emphasizes the need for improved treatments for respiratory insufficiency and respiratory failure [1,2,3]. Currently, the standard of care is mechanical ventilation, which involves invasive procedures that carry significant risks, including barotrauma, ventilator-associated pneumonia, and other infections [4,5]. Furthermore, the complications linked to administering mechanical ventilation to manage novel pandemic illnesses such as COVID-19 have escalated the significance of extracorporeal membrane oxygenation (ECMO), a treatment method that uses an external circuit to circulate blood for gas exchange in an artificial lung, enabling the possibility of reducing or, in rare cases, circumventing the requirement for ventilator support [6,7,8,9].

Microfluidic-based devices with high-permeability membranes could revolutionize the extracorporeal membrane oxygenation (ECMO) process by reducing the blood volume, blood contacting surface area, and overall device size [10,11,12,13,14]. Reduced blood volume is helpful to all patients and critical for smaller patients, such as neonates, as the priming volume of conventional medical devices sometimes exceeds the total blood volume of a neonate [15,16,17]. Additionally, the associated decrease in membrane surface area and blood-contacting artificial materials reduces complications and blood damage [18,19]. Conventional oxygenators are typically a tube-in-tube configuration [20,21], whereas microfluidic devices are often arranged in stacked planar plates to achieve clinically relevant flow rates [22,23,24]. The planar geometry lends itself to a family of recently developed ultra-high-permeability membranes, which have a permeability 10-fold greater than PDMS [25,26], potentially enabling operation with an even smaller surface area and smaller blood volumes than existing microfluidic oxygenator devices. One potential adverse effect of further reducing size is that flow velocities and resulting shear will be larger for a given flow rate.

Acute hemolysis is rare in current ECMO treatments, but effects can aggregate and accumulate with frequent or continuous treatment [27]. At physiologically relevant flow rates, the small dimensions and sharp edges of microfluidic devices may result in shear sufficient to induce hemolysis (rupture of red blood cells) or other bleeding disorders [28,29,30,31]. Many studies have applied empirical and computational models of hemolysis to other blood contacting medical devices, such as valves and blood pumps [32,33,34]. There have also been studies that use microfluidics as a tool to detect hemolytic properties of blood samples [35,36], but there is a lack of studies that delve into the hemolysis of blood-contacting microfluidic medical devices, particularly at the flow rates required for use as an oxygenator.

Potkay et al. (2014) offers a valuable summary of microfluidic device dimensions and performance characteristics prevalent a decade ago [37]. Table 1 highlights critical dimensions and shear rate ranges of the most recent and highest flow rate microfluidic ECMOs reported by three leading research groups within the field, all of which use PDMS membranes. In the table, the maximum wall shear stress within the device was reported. For consistency, we converted the reported maximum wall shear stress values for each group to shear rate [29] by dividing the wall shear stress by the viscosity of blood (0.003 Pa.s), which was either mentioned in the referenced publications [38,39] or assumed based on the studied species if not explicitly provided.

The geometries analyzed in this study are representative microfluidic ECMO devices with micrometer-to-millimeter scale channel widths and flow rates of 100 s of mL min^−1^, which are substantially slower than conventional ECMOs [41], yet are orders of magnitude larger and higher flow than lab-on-chip microfluidics [42,43]. We selected a set of device variants representative of a single module that would incorporate an ultra-high-permeability membrane. The highest shear rates are larger than the devices summarized in Table 1, so our range investigated is inclusive of these devices but extends to a range of possible future devices that might incorporate an ultra-high-permeability membrane or mixing elements, for which the design community has limited experience predicting the device-induced hemolysis (Figure 1).

Computational modeling is commonly employed during the design process of blood-contacting medical devices [49,50]. Hemolysis models are frequently integrated directly into the framework of a computational fluid dynamic (CFD) solver, and these methods guide the design process for devices such as microfluidics [51,52,53], pumps, valves, and catheters [43,44,45]. A range of hemolysis models exist that vary in complexity, but most predict hemolysis based on empirically determined functions of shear stress *τ_s_* and exposure time t, both of which are calculated by CFD [54]. Two methods of implementing hemolysis models are commonly used: (1) power law models, where the damage is a result of the product of time and shear, and (2) time history models, extensions of the power law model, enable the incorporation of shear history effects on RBCs [55]. This allows for the modeling of pre-existing damage, a crucial aspect in understanding the behavior of RBCs under repeated shear stress. Unlike conventional power law models, these models account for the “memory” of past exposures through the inclusion of internal variables that track accumulated damage. This refined approach leads to more accurate predictions of hemoglobin release under subsequent shear exposures. Either of these models is often used to predict the effect of an incremental design variation [56,57]. It is unclear whether existing hemolysis models for the conventional fluidic devices are suitable for predicting hemolysis for the development of micro-oxygenators and micro-dialyzers, as these devices may have greater or non-uniform shear stress, exposure time, and flow characteristics.

Transport within these devices may be limited by the transport rate within the laminar fluid rather than the membrane permeability itself. Thus, geometrical protrusions have been proposed as means to improve mixing within microfluidic channels (Figure 2b). By introducing mixing in the blood side of the microfluidic channel, oxygen-depleted red blood cells (RBCs) can be directed towards the membrane through a flow perpendicular to the membrane surface. This reduces the distance oxygen molecules need to travel within the blood to reach the oxygen-depleted RBCs, promoting faster diffusion and higher saturation rates, ultimately improving overall oxygen transfer efficiency [58,59,60]. The staggered herringbone mixer is a well-characterized mixing element that has been demonstrated to improve mixing for a wide range of flow regimes (1 < Re < 100) [61]. However, the extent to which these flow disturbances caused by herringbone mixers contribute to flow-induced hemolysis has not yet been studied.

The purpose of our study is to assess the accuracy of commonly used power law models in predicting blood damage in representative microfluidic device prototypes. In this study, four geometric variants of a prototype microfluidic ECMO were investigated both computationally and empirically to assess (1) the computational model’s ability to predict hemolysis for different constant parameters and (2) the hemolytic potential of these representative devices. The comparison between the computational and empirical results will allow us to recommend the most suitable power law constant parameters for specific shear stress conditions induced by microfluidic ECMO design variants or more generally for blood-contacting microfluidic medical devices. The design variants intentionally include geometries that we think will cause high shear and possibly hemolysis. In this way, they are laboratory test devices intended to represent the borders of the design space of a clinical device, rather than clinically appropriate devices themselves.

## 2. Materials and Methods

### 2.1. Geometry

We modeled and measured hemolysis in four geometrical variants of a microfluidic oxygenator, as well as positive and negative controls, to explore the model’s predictive power of both absolute magnitude and sensitivity to geometry variations (Figure 3 and Table 2). All the devices contain an inlet manifold that transitions from a circular inlet to rectangular channels that are exposed to the membrane. The fluid in these rectangular channels subsequently flows into an outlet manifold with a circular port for the outlet. The four geometric variations were designed to create variations in both average and maximum shear rates (Appendix A), suggesting the possibility of distinct hemolytic potentials for each device. Exaggerated features that lead to “high shear rates” (15,000 1/s), including a decreased diameter inlet port, reduced channel height, and the presence of herringbones are included to challenge the model’s ability to characterize the effect of these flow features. A device-free circuit (1.6 mm ID tubing) was included as a negative control and a tubing with a 0.4 mm ID was chosen as a positive control, as it generates the highest shear levels among all tested devices while remaining within the observed range of shear experienced by blood-contacting devices.

The microfluidic geometries employed in this study were not designed to represent our best attempt of a clinically effective microfluidic ECMO. Instead, the focus was on creating a diverse range of shear environments within the microfluidic device to induce hemolysis. The 5-channel geometry utilized a moderate channel height with port IDs that matched the loop tubing IDs. This design served as a baseline for comparison. Other devices incorporated different geometries specifically chosen to generate distinct shear rate profiles, theoretically resulting in varying degrees of hemolysis. The herringbone design, for instance, was included to investigate the impact of introducing geometric features within the microfluidic channels on hemolysis. Negative and positive controls were selected to represent the lowest and highest shear levels, respectively. Notably, the range of shear rates achieved across the four test devices and two controls encompassed or surpassed those reported for microfluidic ECMO systems in the literature. This approach of exceeding the anticipated shear conditions served two purposes. First, it allowed for the investigation of hemolysis under potentially harsher environments. Second, by demonstrating negligible hemolysis even under these extreme conditions, the study provides greater confidence in the biocompatibility of devices operating within the expected lower shear range.

### 2.2. Computational Model Set-Up

#### 2.2.1. Numerical Mesh

For each geometry (four device variants and two controls), the fluid path was computationally modeled. Figure 4 shows the fluid path geometry of the baseline device. For each geometry, the solution was resolved for three mesh sizes: 80, 113, and 160 μm CutCell (hexahedral) elements (fine, medium, and coarse, respectively). Mesh refinement regions proportional to the base mesh size were specified for areas with high-velocity gradients to improve solution accuracy and convergence. (Figure 5) The shallow channel (250 μm) of one of the test devices (the reduced gap device) required an element size that was even more refined (57 μm) to resolve flow and reach a converged solution. Likewise, the smaller scale of the positive control required the size of the elements to be reduced to 15 μm. Each mesh contained approximately 24 M hexahedral computational cells at the finest resolution. The average cell orthogonal quality was 0.992.

A mesh convergence study was performed using Richardson extrapolation to assess mesh discretization error [62,63]. For the baseline configuration, the high convergence order suggests that grid-induced error is less than 5%. The results of the mesh convergence study can be seen in Figure 6. Given the order of magnitude of the differences in hemolysis, 5% grid-induced error is considered acceptable.

#### 2.2.2. Computational Algorithm

A computational model of hemolysis using the native fluid flow solver of ANSYS Fluent (2021 R1) and additional User Defined Functions (UDFs) was used. The simulation workflow for the hemolysis is shown in Figure 7. The core CFD algorithm solved for the fluid dynamics of the system, i.e., blood [64,65,66], and the UDF solved for the hemoglobin released and the free hemoglobin. Cell-free hemoglobin (CFH) was defined as a species. A UDF was implemented to model hemolysis by using a source term to create CFH as a function of shear stress and exposure time, as described in greater detail later. In the first box of the hemolysis model flow chart, shear is calculated in three different ways in Ansys fluent. In the box below that, the power law (PL) OR the time history (TH) model equation is set up. For each equation, three different constant parameter sets of C, α, and β were implemented. From this step, cell-free hemoglobin (CFH) was calculated. For the PL, a total of 9 output CFH is achieved. Calculation: (3 shears from ANSYS Fluent) × (3 constant parameter sets) = 9 CFH outputs. Similarly, 9 different CFH outputs are achieved from the TH model.

A parabolic velocity profile (corresponding to either 100 mL min^−1^ or 10 mL min^−1^) was specified at inlets to avoid non-physical wall shear stress. CFH at the inlet of the device was set to zero. Single-pass damage was quantified by the mass-averaged CFH concentration at the outlet.

The solutions from the simulations were attained at steady-state using a blended first- and second-order upwind scheme (blending factor of 0.75). A convergence criterion of 0.5 × 10^−6^ was specified for hemolysis. Flow and hemolysis equations were solved simultaneously on a four-node cluster (4 cores, 32 GB memory per node). The computational time to obtain the solution was approximately 24 h at the finest mesh resolution.

#### 2.2.3. Hemolysis Model

For each iteration, scalar shear stress was calculated in each computational element. The concentration of a species, in this case hemoglobin, is a scalar value, for which the concentration can be solved at every computational cell in the domain according to (Equation (1)), where to *ϕ_k_* is the transported scalar quantity.
(1)$$\frac{\partial }{\partial x_{i}}\left(\rho u_{i}\phi_{k}-\Gamma \frac{\partial \phi_{k}}{\partial x_{i}}\right)={S_{\phi }}_{k}$$

Here, *ρ* is the fluid density, *u_i_* is the velocity of flow along the axial direction, *ϕ_k_* is the transported scalar quantity, and Γ is the diffusion coefficient. The first term *ρu_i_ ϕ_k_* represents the advective term, and the Γ (*∂ϕ_k_*)/(*∂x_i_*) term represents convection. The term on the RHS is the source term, which models the release of free hemoglobin from within cells to the plasma. The source term for each PL model and TH model was defined by the governing Equations (2) and (3), respectively.
(2)$$D(\tau t)=\frac{∆fHb}{Hb}=C\;\tau^{\alpha }\;t^{\beta }$$
(3)$$D(\tau t)=\frac{∆fHb}{Hb}=\sum_{inlet}^{outlet}\;\;C\beta {\left[\;\sum_{\;j=1}^{i}\;\;{\tau_{t_{j}}}^{\frac{\alpha }{\beta }}\;∆t_{j}\;\right]}^{\beta -1}\;{\tau_{t_{i}}}^{\frac{\alpha }{\beta }}\;∆t_{i}$$

The empirical constants (*C*, *α*, and *β*), implemented in the models (PL and TH) were published by Giersiepen, Heuser/Optiz, and Zhang [65,66,67]. Three different groups provided three sets of empirical constants (*C*, *α*, and *β*). The three different methods of computing the scalar shear stress based on the viscous stress tensor reported by Fluent were employed. Two out of the three methods are based on the second stress invariant (*τ*_*v**m*_ and *τ**_p_*), and one is based on the viscous stress tensor (*τ**_b_*). *τ**_p_* is the square root of the absolute value of the second stress invariant of the deviatoric stress tensor [68]. *τ*_*v**m*_ is the von mises stress of this stress tensor, or simply √3**τ*_*p*_. *τ**_b_* is computed from the off-diagonal components of this stress tensor *τ**_b_* as shown in Bludszuweit et al. [69]. A total of eighteen variants of the hemolysis model were studied (PL and TH, 2 * 3 empirical constant sets * 3 scalar stress for each = 18) (Table 3). The values of the constants *C*, *α*, and *β* are listed in Appendix A.

The hemolysis per pass through the device was calculated using Equation (4) as the average index of hemolysis percentage (*IH*%) at the device outlet. *Ht* denotes the hematocrit level in whole blood.
(4)$$IH\%=\frac{∆fHb}{Hb}\;\left(1-Ht\right)\times 100$$

### 2.3. Empirical Experiment Set-Up

The devices were fabricated by stereolithography using Biomed Amber Resin (FormLabs, Somerville, MA, USA) a biocompatible and reusable SLA resin commonly employed for microfluidic applications, using a Form 3 SLA Printer (Formlabs). Induced hemolysis was measured using a recirculating circuit driven by a custom syringe pump (Figure 8a,b). Empirical experiment setups (test circuits) were constructed using 60 mL syringes (Qosina, Ronkonkoma, NY, USA) attached to a custom-built syringe pump, polycarbonate fittings (VWR, Radnor, PA, USA), silicone tubing (MasterFlex, Gelsenkirchen, Germany), microfluidic devices, and blood-bags (100 mL, 2 port blood bag, Qosina).

Citrated bovine calf whole blood was acquired through venipuncture (Lampire Biological Laboratories, Pipersville, PA, USA) from a single donor animal, shipped overnight in a refrigerated container, and used within 12 h of receipt to maintain optimal freshness during experimentation. Anticoagulation with Sodium Citrate was employed by the blood vendor (Lampire) before shipping, to effectively prevent any blood clotting or red blood cell (RBC) aggregation within the microfluidic channels. Blood was filtered through a 250-micron pore mesh (McMaster Carr, Elmhurst, IL, USA). Before each experiment, the circuits were filled with 1 × phosphate-buffered saline (PBS) for priming and circulated for 20 min according to standardized hemolysis testing protocols [70]. The PBS was subsequently removed, and 60 mL of blood was immediately infused into the circuits via the syringe pump. The syringes were positioned vertically, which has two advantages. First, any air that should enter the syringe is not returned to the tubing during the ejection stroke, ensuring that no air bubbles are present within the microfluidic channels. Secondly, this orientation prevents red blood cells from settling in a non-flowing region of the syringe, thus maintaining a consistent hematocrit. The syringes were programmed to pass blood 650 times through each device at a flow rate of 100 mL min^−1^ for both aspiration and infusion. After every 100 strokes of the syringe, a blood sample of 2 mL was aspirated from each circuit. The samples were centrifuged at 13,000× *g* for 10 min and the separated plasma was isolated and centrifuged again to remove any remaining cells. CFH in the plasma was measured according to the Cripps method [63], which is a well-established method for measuring hemolysis that uses light absorbance values at 560, 576, and 593 nm wavelengths by a spectrophotometer (Spectramax iD3, Molecular Devices, San Jose, CA, USA) (Software: SoftMax^®^ Pro 7.0). The index of hemolysis (IH) was computed by accounting for the hematocrit. The number of passes was calculated with consideration of circuit volume, flow rate, and total number of syringe strokes. A total of 4 to 7 biological replicates were collected for each device from distinct donor animals. The mean IH was calculated for each device incorporating hemolysis values obtained from all the days of experiments. The standard mean of error was also calculated for each mean IH. As per the Biomed Amber Resin datasheet, thorough cleaning with 70% Isopropyl alcohol allows for safe and effective device reuse [71].

### 2.4. Data Analysis

The slope of a linear regression between IH% and the number of passes through the device or control (IH% per pass, written as IHPP, index of hemolysis per pass) was used as the metric for damage. IHPP from each of the eighteen variations of the hemolysis model was compared to each other and to corresponding empirical data. The cross-device hemolysis comparison for the empirical experiments was performed using the one-way analysis of variance (ANOVA), where *p* < 0.05 indicates that two sets of data are distinct. The computational and empirical data for the devices were compared by one-sample *t*-test, where a *p* < 0.05 indicates that the computational datum is distinct from the empirical data.

## 3. Results

### 3.1. Simulation: Hemolysis

Simulation results of the 18 different hemolysis model variants for baseline geometry showed little difference between PL and TH models (Figure 9a). Being so, only the PL model variants are shown in Figure 9b,c and hereafter.

The magnitude of hemolysis predicted by the various PL model variant for a single geometry varies drastically (Figure 9b,c), with an overall range of two orders of magnitude difference between PL 3 and 5 (Figure 9b,c). The strongest effect is a result of the parameters C, α, and β derived from the literature, with Giersiepen (PL 1–3) predicting the highest hemolysis and Heuser/Opitz (PL 4–6) the lowest.

The method of calculating shear has a weaker effect on the hemolysis prediction with *τ_p_* shear (PL 3, 6, and 9) consistently predicting four-times-greater hemolysis per pass than the *τ_b_* shear (PL 2, 5, and 8). All model variants are consistent in predicting the relative changes between devices, such that the herringbones increase hemolysis per pass through the device < 10% and all model variants predict the small port and reduced channel gap devices (high shear) induce a 10-to-20-fold increase in hemolysis compared to the featureless device (baseline). The right-most group in Figure 6b depicts the hemolysis values found from the empirical experiments, which will be described in Section 3.3. Empirical vs. Simulation Hemolysis. The negative control showed the lowest level of hemolysis in all model variants when compared to the devices. The positive control had orders of magnitude higher hemolysis compared to the other devices.

### 3.2. Empirical: Hemolysis

Figure 10a shows representative data from one day of empirical experiments. Hemolysis is nearly linear with the number of passes. It is evident that all devices exhibit significantly lower levels of hemolysis (<1.5 × 10^−4^ IHPP) compared to the positive control (~1 × 10^−2^ IHPP). Figure 10b presents the mean hemolysis values of all experiment days. It was observed that even without a device attached to the circuit (tubing only), the mean hemolysis values were found to be statistically comparable to the values that which the devices were attached (*p* > 0.05). This suggests the device-induced hemolysis (excluding pump, tubing, and connectors), calculated as the difference between the maximum device-induced damage and the tubing-only circuit is 1.5 × 10^−4^ IHPP (3 × 10^−4^–1.5 × 10^−4^ = 1.5 × 10^−4^ IHPP). As the empirical hemolysis values for 100 mL min^−1^ flow rates were so low, we determined that 10 mL min^−1^ flow conditions would generate hemolysis well below our measurement sensitivity; thus, empirical experiments were not performed for 10 mL min^−1^. While comparing within the empirical cases, the baseline hemolysis value was not statistically distinguishable from the other devices (herringbone, reduced port, and reduced gap) (*p* > 0.05). The positive control IHPP values on the other hand were statistically significantly different from all other devices (*p* < 0.05). Appendix A reports the cross-device comparison *p* values obtained from the one-way ANOVA test.

### 3.3. Empirical vs. Simulation Hemolysis

For all cases (devices and controls) PL 1–3 overpredicted (IHPP to IHPP) the hemolysis values when compared to the empirical (Figure 11). PL 4–6 underpredicted (to IHPP) for the negative control, baseline, and herringbone device, but the prediction they provided for relatively higher shear devices, namely reduced port, reduced gap, and positive control, were rather accurate. Likewise, PL 7 and 8, which best agreed for the negative control, baseline, and herringbone device, overpredicted (to 0.45 IHPP) for reduced port, reduced gap, and positive control. PL 9 followed a similar trend as PL 7 and 8 with lower accuracy. Appendix A reports the empirical vs. simulation comparison *p* values obtained from one sample *t*-test.

## 4. Discussion

Our results indicate that the time history model did not significantly improve the prediction accuracy compared to the simpler PL model. This observation aligns with our expectations. Time history models account for the cumulative effect of shear stress exposure on red blood cells (RBCs). When the predicted hemolysis levels are low, as in our study, the additional stress from past exposures likely plays a minimal role in further damaging the RBCs. In simpler terms, since the overall predicted hemolysis was low, the additional stress from the RBC history was not significant enough to noticeably affect the final prediction.

The PL model variants predicted hemolysis values across several orders of magnitude depending on which set of empirical parameters was used. This is discouraging as the ideal model would be one that accurately predicts the absolute value of hemolysis in any device. Unfortunately, consensus on this has not yet been reached, even in the field of blood pumps, in which similar hemolysis models have been applied for a long time. Significant variations exist in the shear rate ranges and stress application methods employed for model development. These discrepancies can lead to weakened correlations between predicted hemolysis and the actual behavior of specific devices, particularly in microfluidics. As observed, the predicted hemolysis can depend on the chosen model, not solely on the microfluidic design.

Despite these limitations, our study clarifies the current model’s capabilities. While it might not capture subtle variations in hemolysis between the tested devices, it successfully identified the high hemolysis induced by the positive control. This suggests the model’s ability to predict hemolysis trends for design iterations, provided the predicted hemolysis surpasses a certain threshold (IHPP)—currently defined based on the positive control data. As discussed later in more detail, future work will focus on refining the model for improved hemolysis prediction across a broader range of shear stress conditions and device geometries. This will enhance researchers’ confidence in utilizing the model for microfluidic oxygenator design and evaluation.

Our observations revealed that for devices with average shear rates below 500 s^−1^ (baseline, herringbone, and negative control) (Appendix A), PL 7 and 8 exhibited the closest predictions. Conversely, for devices operating at shear rates equal to or exceeding 600 s^−1^ (reduced port, reduced gap, and positive control) (Appendix A), PL 4–6 demonstrated the closest predictions. This relationship suggests that when designing a microfluidic device expected to operate in a low shear range, selecting PL 7 and 8 for hemolysis prediction is optimal. Conversely, when the prototype device is anticipated to operate in a high shear range, selecting PL 4–6 is ideal. Although beyond the scope of this work, recent studies have presented methods to develop device-specific parameters using empirical data that span multiple flow rates and a range of device geometries [72]. They determined device-specific model parameters, through a high number of empirical replicates and Kriging surrogate modeling, that produce hemolysis values that are closer to the empirical findings.

The coefficients for the Giersiepen model (PL 1–3) were derived from experiments encompassing shear rate ranges from 0 s^−1^ to 70,833 s^−1^, for the Heuser/Optiz model (PL 4–6) the shear rate ranged from 10,000 s^−1^ to 166,700 s^−1^, and for the Zhang model (PL 7–9), the shear rate ranged from 13,890 s^−1^ to 88,890 s^−1^. Notably, the Heuser/Optiz model variants exhibited the highest shear rate ranges while calculating their coefficients, and as mentioned earlier, these models demonstrated good agreement with high-shear devices. This observation might suggest that the coefficients determined by Heuser/Optiz, being derived from shear rate ranges closest to those employed by high-shear devices, resulted in a closer agreement with these devices. Conversely, for the other model variants, such a clear correlation is not evident. This lack of correlation might be attributed to the differences in shear application methods utilized by those models compared to our approach of using microfluidics. Other factors, such as differing blood characteristics may also contribute to differences between empirical data and the previously derived models. In summary, the variations in the experimental shear rate ranges and the methods of shear application among different models can potentially impact their correlation with specific devices, especially in the context of microfluidics-based studies. A future work could be to recommend new sets of coefficients derived from microfluidic shear applications, which can lead to the development of more accurate hemolysis prediction models for microfluidics.

We acknowledge the following limitations inherent to our current model. Model Complexity: The employed model utilizes a simplified approach to hemolysis prediction by directly correlating shear stress and exposure time with cell-free hemoglobin (CFH) generation. While this approach offers computational efficiency, it may not capture the full complexity of hemolysis, which can be influenced by additional factors like RBC morphology, membrane properties, and the presence of other blood components. Empirical Parameter Dependence: The model relies on empirical parameters (C, α, and β) within the power law (PL) and time history (TH) equations. These parameters were empirically determined by prior authors under conditions that are not identical to those in the devices that we presented in this work. As we demonstrated, there are very large differences in the predictions of the models presented and no consensus on which is most appropriate for these devices. Mesh dependence: Figure 6 demonstrates a clear dependence of predicted hemolysis values on mesh resolution. Coarser meshes exhibited greater estimated error compared to finer meshes, suggesting a potential influence of mesh size on the accuracy of the simulations. We recognize the need to address these limitations in future studies. Exploring more sophisticated hemolysis models that incorporate additional factors beyond shear stress and exposure time could provide a more comprehensive understanding of hemolysis behavior [34,73]. Further validation of the employed empirical parameters across a wider range of experimental conditions and blood sources would strengthen the model’s generalizability and reduce uncertainties in the predictions.

For all model variants, the three device modifications yielded an increase in hemolysis over the baseline device. This means the model variants were sensitive to each of the three design changes introduced and predicted an increase in hemolysis for all three, with the largest incremental increase resulting from the two designs intended to increase shear (reduced port and reduced gap). The model variants may be useful for device design if multiple devices are required to be compared with each other in terms of their hemolytic tendency.

Empirical Data: Empirical hemolysis observed for all four microfluidic devices was remarkably low (<2 ppm hemolysis) and statistically indistinguishable for all four design iterations when compared to the negative control (*p* > 0.05) (Figure 7). This indicates that the bulk of hemolysis is caused by the syringe pump and possibly fittings in tubing. In a prior iteration of this experiment, we used a peristaltic pump, which created substantially more damage than the syringe pump. The low levels of device-induced damage within all devices is promising for clinical use. A maximum of 3 ppm per pass of RBCs are damaged by the device, tubing, and pump combined, which is lower than the amount damaged by conventional ECMO loops (~10 ppm per pass) [74].

The incremental damage caused by any of these devices is less than 1.5 ppm per pass. Although these devices contain sharp edges and small features that induce non-physiological shear stresses at the tested flow rates, these stresses appear insufficient to induce significant hemolysis in the devices. The experimental conditions (flow rate: 100 mL min^−1^; reservoir volume: 60 mL; experiment length: 5–8 h) generate over 650 passes through the devices, which is comparable to the time required for short-term ECMO support as a “bridge” [75]. The number of passes through the device in these experiments may greatly exceed what would normally be experienced if used clinically as a wearable continuous-flow device.

As previously stated, the results of empirical hemolysis for all four devices did not demonstrate a significant difference when compared to the negative control. This suggests that the hemolysis induced by the test loop and syringe pump may have overshadowed any hemolysis induced by the devices themselves. Therefore, the device-induced hemolysis level was deemed negligible for the tested devices. However, it is important to note that the positive control hemolysis was considerably higher, as confirmed by both empirical and computational analyses. This result indicates that even in high shear generated by devices such as reduced gap (~105,000 1/s) and reduced port (~150,000 1/s), significant hemolysis is not observed. While the existing literature on microfluidic ECMO reports shear rates typically around 1000 1/s to 2000 1/s (Table 1), our devices were designed to achieve significantly higher single-channel flow rates, velocities, and consequently, shear rates, that might accompany ultra-high-permeability membranes with lower surface areas. This suggests the hemolysis observed in the literature devices (devices in Table 1) will be negligible. This finding extends to a broader range of device geometries and operating conditions, suggesting that the risk of hemolysis induced by shear stress within microfluidic ECMO devices remains low even under harsher conditions. This provides future researchers with greater design freedom when exploring novel microfluidic concepts, as long as anticipated shear rates fall within or below the ranges investigated in this study. The selection of a narrow tube as a positive control (~273,000 1/s) served a critical purpose. By establishing a scenario with demonstrably high hemolysis levels, we validated our chosen assessment method. This confirms that the employed techniques are capable of detecting very small levels of hemolysis, but also just that the devices caused effectively negligible levels of hemolysis.

Although the computational model did not accurately predict subtle differences in hemolysis between the devices, they were able to differentiate high levels of hemolysis induced by the positive control (~273,000 s^−1^). This indicates that for microfluidic design iterations, the model can predict the trend of hemolysis accurately if the modeled hemolysis exceeds a certain threshold IHPP. Based on our experiments, this threshold is currently set at the hemolysis level obtained from the positive control. In future studies, it will be necessary to refine this threshold value and determine the lowest IHPP for computational analyses to align with empirical hemolysis measurements. Furthermore, future experiments should consider incorporating both computational and empirical testing of thrombosis for the devices to enhance hemocompatibility assessment.

The use of multi-functional or multi-physical models to guide device design is promising and increasingly practical with advancements in techniques and computer power. The ability to study design variants through simulation leads to less expensive and faster design iteration, eventually resulting in better-optimized designs. The low device-induced hemolysis (<2 ppm) measured in our study at physiologically relevant flow rates is promising for the future development of microfluidic dialyzers and oxygenators, and our work demonstrates that computational modeling may supplement empirical testing to expedite the design optimization of these devices. Further, this study indicates that the PL has a higher tendency to over-predict hemolysis in microfluidic devices, and further customization is required to implement these existing model variants in a microfluidic device geometry.

## Figures and Tables

**Figure 1 micromachines-15-00790-f001:**
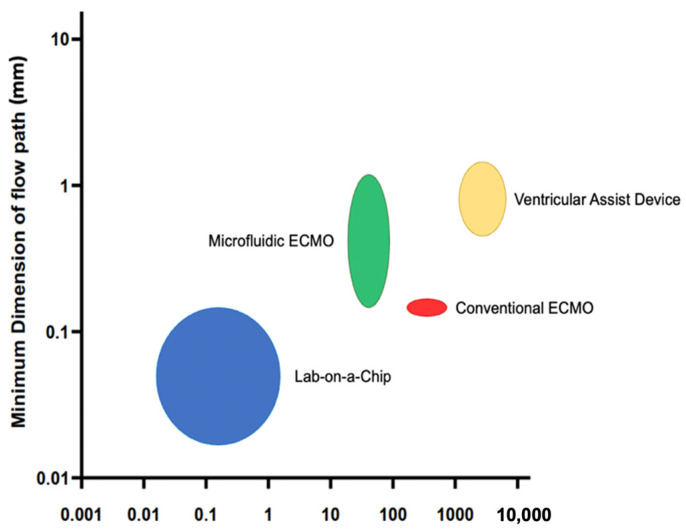
Comparison of different types of blood-contacting medical devices; lab-on-a-Chip, microfluidic ECMO [44,45], conventional ECMO [41,46,47], and ventricular assist device [48] in terms of minimum dimension of blood flow path and flow rate through the channel of minimum dimension. The height and width of the circle/oval represent the minimum dimension and flow rate ranges, respectively.

**Figure 2 micromachines-15-00790-f002:**
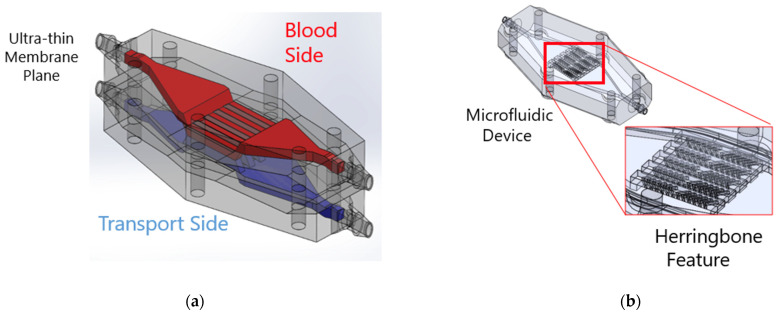
(**a**) Microfluidic ECMO with ultra-thin highly permeable membrane, (**b**) blood side of the ECMO with herringbone mixers.

**Figure 3 micromachines-15-00790-f003:**
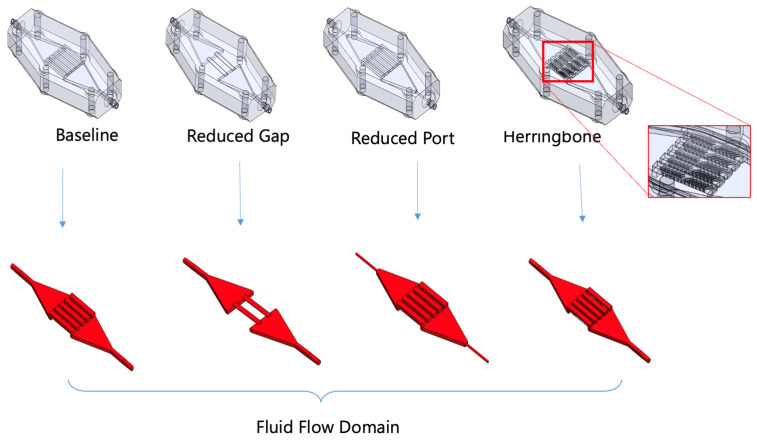
Device variants and their corresponding flow domains. Baseline: five open channels; Reduced Gap: fewer channels with the smallest height to increase shear; Reduced Port: small inlet and outlet ports; Herringbone: channels with herringbone features to enhance mixing; herringbone height is half the channel height.

**Figure 4 micromachines-15-00790-f004:**
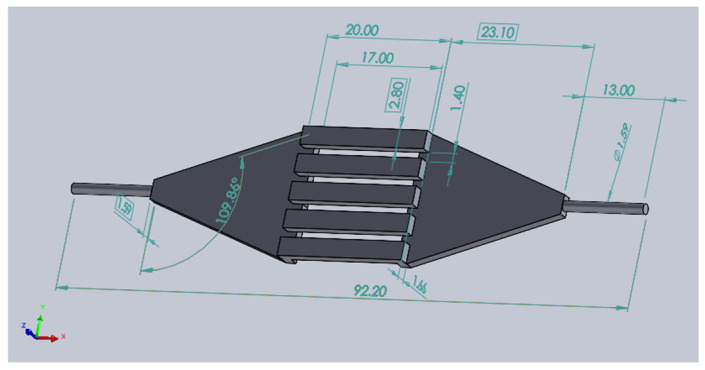
Fluid path of the baseline geometry with dimensions in mm.

**Figure 5 micromachines-15-00790-f005:**
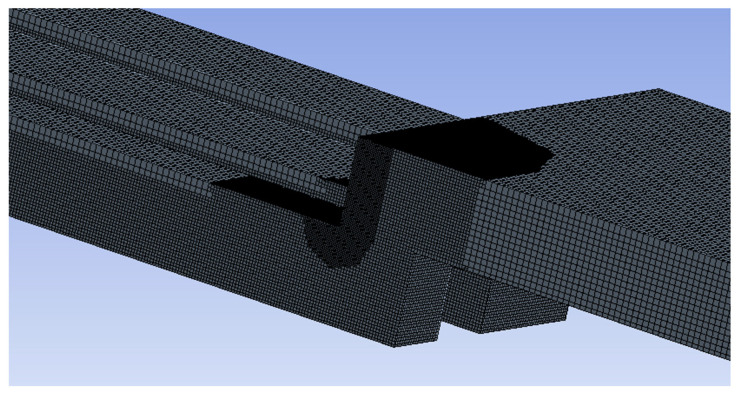
Detail from the medium-level mesh on baseline geometry.

**Figure 6 micromachines-15-00790-f006:**
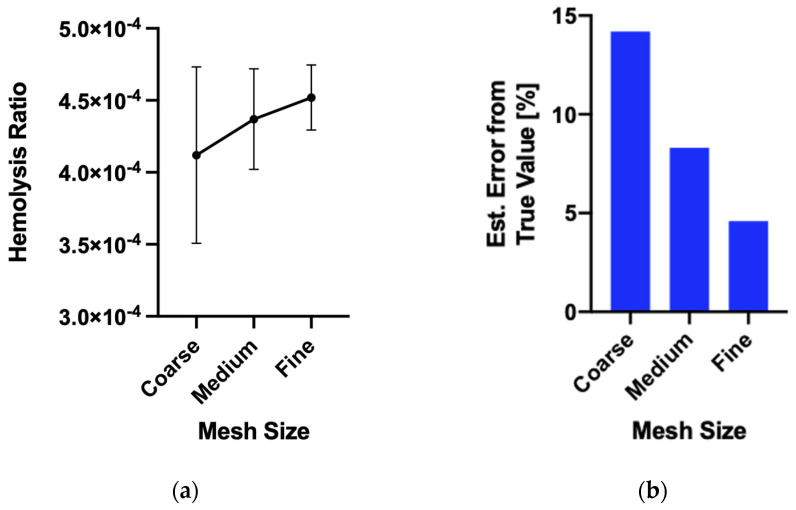
(**a**) The predicted hemolysis increased with finer mesh. (**b**) The estimated error from true value decreased as the mesh was made finer. Error calculated using Richardson extrapolation for baseline case.

**Figure 7 micromachines-15-00790-f007:**
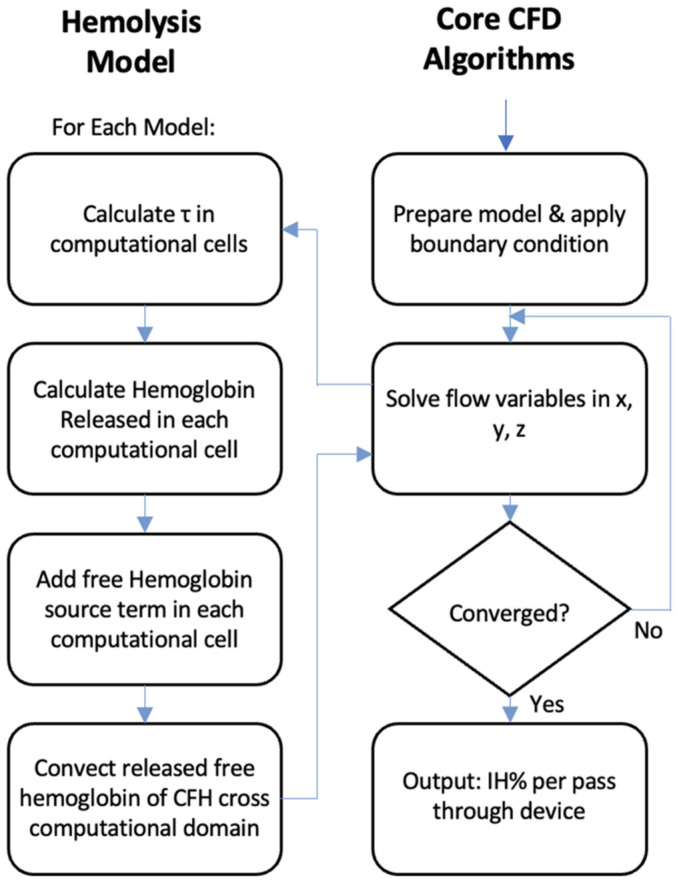
Flow chart of the hemolysis incorporated via UDFs. Fluent solves for fluid flow using the core algorithms (illustrated in the right column). Hemoglobin is treated as a species with a source term that allows for the generation of hemolysis from shear stresses.

**Figure 8 micromachines-15-00790-f008:**
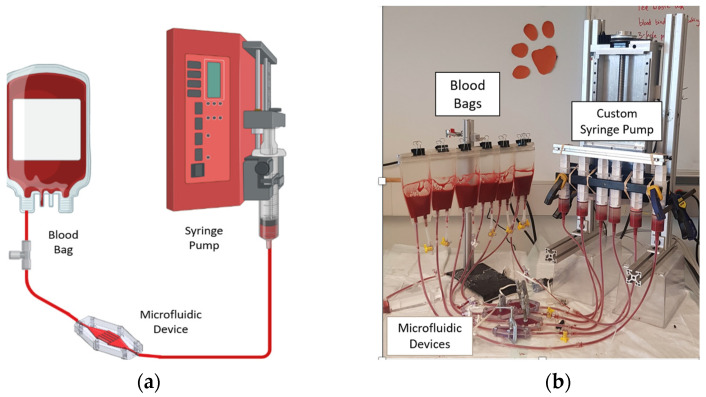
(**a**) Schematic diagram of the experimental setup. (**b**) Test circuit with custom syringe pump, microfluidic devices, and blood bags.

**Figure 9 micromachines-15-00790-f009:**
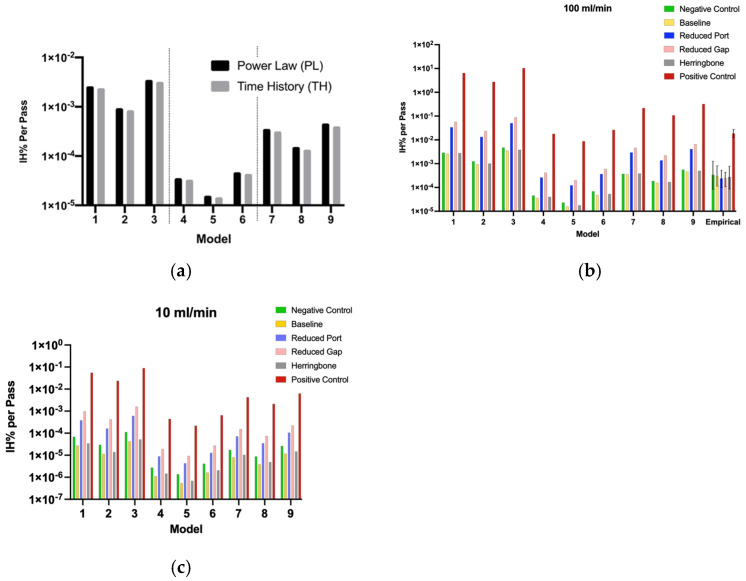
(**a**) Comparison of hemolysis data for the baseline device for both PL and TH models (total 18 variants) with Giersiepen parameters (PL 1–3), Heuser/Opitz parameters (PL 4–6), and the Zhang parameters (PL 7–9); (**b**,**c**) simulation hemolysis values for the power law model variants for all four devices and controls for flow rates 100 mL min^−1^ (**b**) and 10 mL min^−1^ (**c**), respectively. Panel B also includes the empirical hemolysis values.

**Figure 10 micromachines-15-00790-f010:**
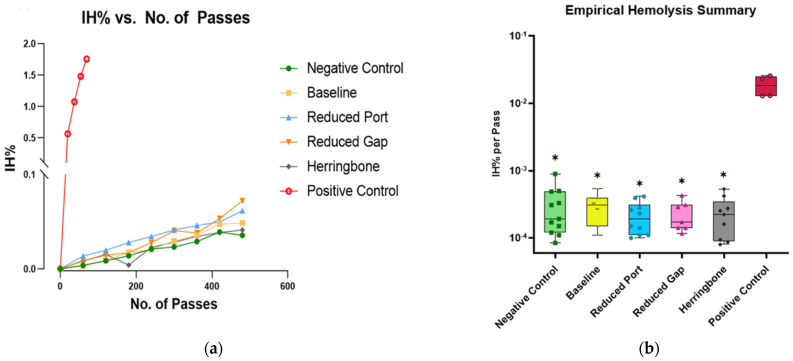
(**a**) IH% Vs. No. of Passes plotted for one empirical experiment. (**b**) Overall mean of empirical data comparing IHPP for each device and controls. The solid midline represents the median IHPP and the whiskers are the lower and upper range of IHPP. Boxes containing asterisks depicts significantly different IHPP value from the positive control (*p* < 0.005).

**Figure 11 micromachines-15-00790-f011:**
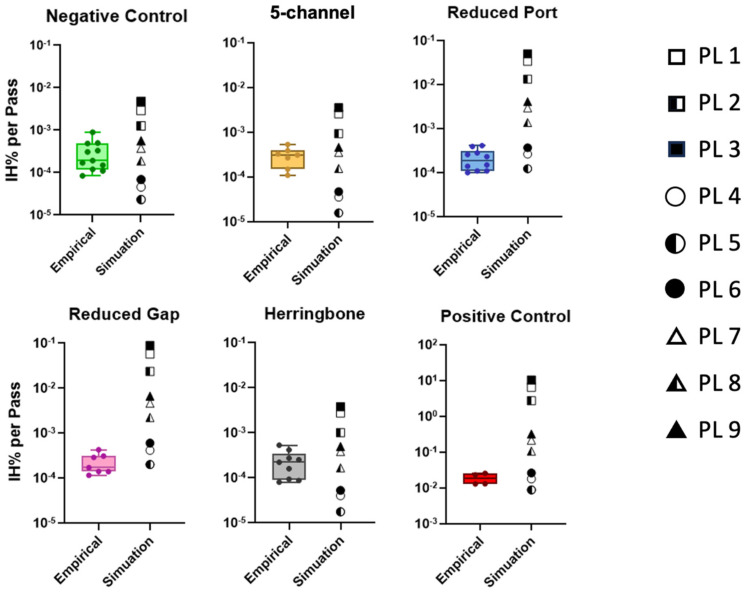
IH% per pass for individual devices and controls: empirical vs. simulation. The whiskers are the lower and upper range of IHPP. The squares, circles, and triangles represent Giersiepen (PL 1 to 3), Heuser/Optiz (PL 4 to 6), and Zhang (PL 7 to 9) PL model variants, respectively.

**Table 1 micromachines-15-00790-t001:** Geometrical features and shear rate ranges of microfluidic ECMO devices from leading research groups: Borenstein [38,39], Potkay [40], and Selvaganapathy [12].

Research Group	Channel Dimension			Nominal Device Flow Rate (mL/min)	Flow Rate per Layer (mL/min)	Single Channel Flow Rate (mL/min)	Max. Wall Shear Stress (Pa)	Calculated Shear Rate (1/s)
	Height (µm)	Width (µm)	Length (mm)					
Borenstein [38,39]	160	500	91	750	94 **	0.136	3.5	1170
Potkay [40]	200	297	7.3	6	6	0.03	3.3	1100
Selvaganapathy [12]	160	1000	91	60	7.5 **	1.1 *	6	2000

* Note: For the Selvaganapathy device, the single-channel flow rate was estimated by assuming inlet streamlines represent individual flow paths. ** Note: The Borenstein and Selvaganapathy device comprised multiple repeating units sometimes called layers [38,39], sometimes called units [12]. In this manuscript, we are calling each unit a “layer”.

**Table 2 micromachines-15-00790-t002:** Geometrical features of microfluidic devices. The difference from the baseline device is bolded.

	Baseline	Reduced Port	Reduced Gap	Herringbone	Positive Control	Negative Control
# of Channels	5	5	2	5	−	−
Channel Width (mm)	1.6	1.6	1.6	1.6	−	−
Channel Height (mm)	2.8	2.8	**0.25**	2.8	−	−
Channel Length (mm)	20	20	20	20	30	50
Port ID (device)	1.6	**1**	1.6	1.6	0.4	1.6
# Herringbones/channel	0	0	0	**12**	−	−

**Table 3 micromachines-15-00790-t003:** Hemolysis model parameters [29,56,57,58].

	Shear Stress Variants		
	*τ_vm_*	*τ_b_*	*τ_p_*
Giersiepen	PL-1, TH-1	PL-2, TH-2	PL-3, TH-3
Heuser/Optiz	PL-4, TH-4	PL-5, TH-5	PL-6, TH-6
Zhang	PL-7, TH-7	PL-8, TH-8	PL-9, TH-9

## Data Availability

The original contributions presented in the study are included in the article, further inquiries can be directed to the corresponding author.

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
