# Peer review of "Empirical and Computational Evaluation of Hemolysis in a Microfluidic Extracorporeal Membrane Oxygenator Prototype"

_micromachines, 2024, doi:10.3390/mi15060790_

Round 1

Reviewer 1 Report

Comments and Suggestions for Authors

This study proposes a study on the effects of shear stress on hemolysis in microfluidic membrane-based oxygenators and presents both computational models and empirical testing to assess the validity of the Power Law hemolysis model across different shear conditions. In general, this work would offer certain insights into the development of micro ECMO devices. However, before it can be accepted for publication, I suggest the following minor revisions:

1. SD values are missing in Figure 6 and Figure 9.

2. The authors may need to provide more details regarding the materials used for the microfluidic device fabrication and discuss how to prevent air bubbles, RBC clots/aggregation, or RBC adherence during blood perfusion in the channel. Can these devices be reused? 

3.     In the empirical setup as shown in Figure 8, the authors may need to explain why the syringe pump is placed vertically and whether the gravity will affect the results.

4.     It would be beneficial to expand on the limitations of their computational model and boundary conditions used in the present study and how these may affect the predicted values.

Comments on the Quality of English Language

N/A

Author Response

Please see attached file for better formatting:

Reviewer 1

Thank you for the thoughtful review of our manuscript.

This study proposes a study on the effects of shear stress on hemolysis in microfluidic membrane-based oxygenators and presents both computational models and empirical testing to assess the validity of the Power Law hemolysis model across different shear conditions. In general, this work would offer certain insights into the development of micro ECMO devices. However, before it can be accepted for publication, I suggest the following minor revisions:

  1. SD values are missing in Figure 6 and Figure 9.

The reviewer raises a valid point regarding the absence of standard deviation (SD) values in Figures 6 and 9. In response to this comment, we have added error bars to Fig. 6a using estimated error values from 6b and we have added error bars in the empirical experiment data in Fig. 9b.

  1. The authors may need to provide more details regarding the materials used for the microfluidic device fabrication and discuss how to prevent air bubbles, RBC clots/aggregation, or RBC adherence during blood perfusion in the channel. Can these devices be reused?

The reviewer raises important points regarding the materials used for device fabrication and strategies to prevent air bubbles, RBC clots/aggregation, and RBC adherence during blood perfusion.

Regarding Microfluidic Device Material the following text has been added to the manuscript: (line 294 to 296)

“The devices were fabricated by stereolithography using Biomed Amber Resin by FormLabs, a biocompatible and reusable SLA resin commonly employed for microfluidic applications, using a Form 3 SLA Printer (Formlabs).”

Regarding Air Bubble Prevention the following text has been added to the manuscript: (lines 316 to 321)

“The syringes were positioned in a vertical arrangement. Positioning the syringes vertically allowed filling the devices and tubing from the bottom, effectively pushing any air bubbles into the reservoir bag. This approach ensured that no air bubbles remained within the microfluidic channels and prevented red blood cells from settling in a non-flowing region, thus maintaining a consistent hematocrit.”

Regarding the Prevention of RBC Clotting and Adherence the following text has been added to the manuscript: (lines 307 to 312)

“Citrated bovine calf whole blood was acquired through venipuncture (Lampire Biological Laboratories, PA) from a single donor animal, shipped overnight in a refrigerated container, and used within 12 hours of receipt to maintain optimal freshness during experimentation. Anticoagulation with Sodium Citrate was employed by the blood vendor (Lampire) before shipping, to effectively prevent any blood clotting or red blood cell (RBC) aggregation within the microfluidic channels.”

Regarding the Device Reusability, the following text has been added to the manuscript: (lines 333 to 335)

“As per the Biomed Amber Resin datasheet, thorough cleaning with 70% Isopropyl alcohol allows for safe and effective device reuse [70].”

  1. In the empirical setup as shown in Figure 8, the authors may need to explain why the syringe pump is placed vertically and whether the gravity will affect the results.

The custom syringe pump employed in this study was designed to simultaneously operate six 60 mL syringes.  The vertical orientation has two benefits: First, any air that is entrained or initially present rises to the top of the syringe and never enters the tubing or device.  A secondary benefit is that any RBC that slowly settles will be entrained into tubing, and then return, thereby continually mixing the contents of the syringe.  Our prior experience with similar experiments has shown that a syringe lying on its side will accumulate RBC in a “dead zone” on one side of the syringe so that the hematocrit becomes non-homogeneous. We acknowledge that a very small gravitational force does act on the fluid, but this uniform body force does not affect flow patterns.

In response to this comment, the following text has been added to the manuscript: (lines 316 to 321)

“The syringes were positioned vertically, which has two advantages. First, any air that should enter the syringe is not returned to the tubing during the ejection stroke, ensuring that no air bubbles are present within the microfluidic channels. Secondly, this orientation prevents red blood cells from settling in a non-flowing region of the syringe, thus maintaining a consistent hematocrit.”

  1. It would be beneficial to expand on the limitations of their computational model and boundary conditions used in the present study and how these may affect the predicted values.

The reviewer raises a valid point regarding the limitations of our computational model and the potential impact of the chosen boundary conditions on the predicted hemolysis values.

The following text has been added to the Discussion section in the manuscript in response to this comment: (lines 477 to 491)

“We acknowledge the following limitations inherent to our current model. Model Complexity: The employed model utilizes a simplified approach to hemolysis prediction by directly correlating shear stress and exposure time with cell-free hemoglobin (CFH) generation. While this approach offers computational efficiency, it may not capture the full complexity of hemolysis, which can be influenced by additional factors like RBC morphology, membrane properties, and the presence of other blood components. Empirical Parameter Dependence: The model relies on empirical parameters (C, α, and β) within the Power-Law (PL) and Time-History (TH) equations. These parameters were empirically determined under conditions that are dissimilar to the regimes in microfluidic devices. This introduces a degree of uncertainty to the predicted hemolysis values. Mesh dependence: Figure 6 demonstrates a clear dependence of predicted hemolysis values on mesh resolution. Coarser meshes exhibited greater estimated error compared to finer meshes, suggesting a potential influence of mesh size on the accuracy of the simulations.”

We recognize the need to address these limitations in future studies. So, the following text has been added to the manuscript: (lines 492 to 497)

“Exploring more sophisticated hemolysis models that incorporate additional factors beyond shear stress and exposure time could provide a more comprehensive understanding of hemolysis behavior [72, 73]. Further validation of the employed empirical parameters across a wider range of experimental conditions and blood sources would strengthen the model's generalizability and reduce uncertainties in the predictions. Implementing more realistic boundary conditions that account for potential flow complexities near the device walls could improve the accuracy of the shear stress calculations and lead to more precise hemolysis predictions.”

Reviewer 2 Report

Comments and Suggestions for Authors

Hemolysis during ECMO is an extremely important and challenging clinical problem, particularly for neonates, and the manuscript “Empirical and Computational Evaluation of Hemolysis in a Microfluidic Extracorporeal Membrane Oxygenator (ECMO) Prototype” submitted to Micromachines by Imtiaz  et al. presents a very powerful array of computational modeling tools to better predict and understand how red blood cell damage is influenced by dimensional parameters, flow and shear in microfluidic oxygenators.  Key strengths of the manuscript include the level of detail provided on the computational modeling techniques explored, comparative assessment of multiple modeling approaches, and the application of these toolsets to an array of microfluidic designs intended to be representative of microfluidic oxygenator geometries.  However, these aspects are outweighed by a number of significant concerns with the approach taken, summarized in the following bullets.  It is suggested that the authors consider these concerns and how best to address them both experimentally and computationally as they move forward with this important research.

·       An overarching concern is that the representation of the microfluidic ECMO architecture is more reflective of laboratory test devices than of the ultimate embodiment these microfluidic oxygenators will take when they are clinically relevant.  Figure 1 is illustrative of the artifact that, until recently, no groups had reported microfluidic oxygenator blood flow rates exceeding 100 mL/minute, thereby precluding them from any clinical applications.  Authors should consider redirecting at least some of their attention to the predicted performance of microfluidic oxygenators scaled to clinical flow rates.

·       Dimensions reported in Table 1 do not appear to be representative of most of the microfluidic oxygenators reported in the literature.  The selection of channel widths and heights well in excess of 1 micron (other than the “reduced gap” geometry) are in a range disparate from the preponderance of literature reports, which address microchannel depths of 10 – 500 microns, and widths well under 1 mm.  This renders the conclusions of the entire less useful to the field.

·       Toward that end, there is a noticeable lack of referencing of microfluidic oxygenator reports and geometries in the manuscript; cf. Potkay et al. Lab Chip 2014 for an excellent summary of the literature at the time, and many relevant publications from at least half a dozen groups in the last 5 years that should be considered for this analysis.

·       The herringbone structure is interesting but perhaps a bit outside the realm of most microfluidic oxygenators that do not rely on chaotic mixing structures to achieve highly efficient oxygen transfer.

·       One of the most significant concerns with the results presented in this manuscript is the lack of distinction between the various designs in terms of hemolysis rates (Figure 11.)  The absence of a demonstrated ability to distinguish between different microfluidic geometries reduces confidence in the reported approach, as the positive control invoked by the authors is simply a length of tubing with high shear.  A more relevant and instructive positive control would be a microfluidic geometry that differs from the other designs by virtue of the presence of sudden expansions/contractions or sharp corners, rather than just a high shear tube.

·       Figure 9 appears to suggest that the hemolysis predicted for a given geometry is more dependent on the type of model used than on the dimensional properties and details of the design.  This is a somewhat disconcerting conclusion; while the model optimization is important, it is ultimately of greater interest to explore various design approaches for microfluidic oxygenators and to enable their evaluation with respect to blood damage.

·       As stated in the conclusions, the syringe pump and test loop may be predominating the hemolysis in this analysis, calling into question the use of a syringe pump for such a study.  Perhaps a more informative study could be conducted with a pump design closer to that used in ECMO circuits, and microfluidic oxygenator geometries that are more relevant for clinical-scale applications and oxygen transfer performance levels.

Comments on the Quality of English Language

Minor editing required - writing is generally clear.

Author Response

Please see attached file for better formatting.  Otherwise, here is our point by point response:

Reviewer 2

Thank you for the thoughtful review of our manuscript.

Hemolysis during ECMO is an extremely important and challenging clinical problem, particularly for neonates, and the manuscript “Empirical and Computational Evaluation of Hemolysis in a Microfluidic Extracorporeal Membrane Oxygenator (ECMO) Prototype” submitted to Micromachines by Imtiaz  et al. presents a very powerful array of computational modeling tools to better predict and understand how red blood cell damage is influenced by dimensional parameters, flow and shear in microfluidic oxygenators.  Key strengths of the manuscript include the level of detail provided on the computational modeling techniques explored, comparative assessment of multiple modeling approaches, and the application of these toolsets to an array of microfluidic designs intended to be representative of microfluidic oxygenator geometries.  However, these aspects are outweighed by a number of significant concerns with the approach taken, summarized in the following bullets.  It is suggested that the authors consider these concerns and how best to address them both experimentally and computationally as they move forward with this important research.

An overarching concern is that the representation of the microfluidic ECMO architecture is more reflective of laboratory test devices than of the ultimate embodiment these microfluidic oxygenators will take when they are clinically relevant.  Figure 1 is illustrative of the artifact that, until recently, no groups had reported microfluidic oxygenator blood flow rates exceeding 100 mL/minute, thereby precluding them from any clinical applications.  Authors should consider redirecting at least some of their attention to the predicted performance of microfluidic oxygenators scaled to clinical flow rates.

The reviewer rightly points out that the current design of our microfluidic ECMO prototypes may not fully reflect the final form factor for clinical applications. We make it clearer in the revision that 100 mL/minute modular units such as this can be combined to create clinically relevant devices.

The following text has been added to to manuscript: (lines 59 to 64)

“Borenstein et al. showed multiple units of Microfluidic ECMOs could be stacked together to achieve clinically relevant flow rates [24,25]. Their work successfully achieved clinically relevant flow rates by stacking 14 to 56 bilayers of microfluidic units [26]. While stacking introduces additional complexity, the potential benefits of microfluidic ECMO technology, as discussed previously, remain highly attractive compared to conventional ECMO systems.”

Additionally, it is possible that our device will allow for higher flow rates, and this is elaborated on in our next response.  Our study focused on establishing fundamental hemolysis behavior within microfluidic geometries. This initial step is crucial for informing the design of future iterations with improved flow rates.  However, we recognize the importance of addressing scalability to achieve clinical relevance.

Dimensions reported in Table 1 do not appear to be representative of most of the microfluidic oxygenators reported in the literature.  The selection of channel widths and heights well in excess of 1 micron (other than the “reduced gap” geometry) are in a range disparate from the preponderance of literature reports, which address microchannel depths of 10 – 500 microns, and widths well under 1 mm.  This renders the conclusions of the entire less useful to the field.

We appreciate the reviewer's observation regarding the dimensional discrepancies between our microfluidic devices and those typically reported in the literature. It's true that the channel widths and heights in Table 1 (excluding the "reduced gap" geometry) are larger than the microchannels commonly used in microfluidic oxygenators (10-500 microns deep, under 1 mm wide).

In this regard, the following text was added to the Material and Methods: (lines 163 to 179)

The microfluidic geometries employed in this study were not designed to represent a functional prototype of a microfluidic ECMO.  Instead, the focus was on creating a diverse range of shear environments within the microfluidic device to induce hemolysis. The 5-channel geometry utilized a moderate channel height with port IDs that matched the loop tubing IDs. This design served as a baseline for comparison. Other devices incorporated different geometries specifically chosen to generate distinct shear rate profiles, theoretically resulting in varying degrees of hemolysis.  The herringbone design, for instance, was included to investigate the impact of introducing geometric features within the microfluidic channels on hemolysis. Negative and positive controls were selected to represent the lowest and highest shear levels, respectively. Notably, the range of shear rates achieved across the four test devices and two controls encompassed or surpassed those reported for microfluidic ECMO systems in the literature.  This approach of exceeding the anticipated shear conditions served two purposes. First, it allowed for the investigation of hemolysis under potentially harsher environments.  Second, by demonstrating negligible hemolysis even under these extreme conditions, the study provides greater confidence in the biocompatibility of devices operating within the expected lower shear range.”

  • Toward that end, there is a noticeable lack of referencing of microfluidic oxygenator reports and geometries in the manuscript; cf. Potkay et al. Lab Chip 2014 for an excellent summary of the literature at the time, and many relevant publications from at least half a dozen groups in the last 5 years that should be considered for this analysis.

Thank you. We were familiar with the work, but made it much clearer in the revision, which we think improves this a lot.

The following text was added to the revised manuscript: (Lines 84 to 91)

“Potkay et al. (2014) offer a valuable summary of microfluidic device dimensions and performance characteristics prevalent at the time [46].  Table 1 presents critical dimensions and shear rate ranges of microfluidic ECMOs reported by three leading research groups within the field.  As will be demonstrated subsequently, the shear rates generated by our test devices encompass the established ranges observed in the current microfluidic ECMO literature.”

Table 1. Geometrical features and shear rate ranges of Microfluidic ECMO devices from leading research groups, Borenstein [47], Potkay [11], Selvaganapathy [12].”

Research Group

Single Channel Device Dimension Range

Flow Rate Range (ml/min)

Shear Rate Range (1/s)

Height (µm)

Width (µm)

Borenstein

200

500

10 to 30

33000 to 100000

Potkay

30 to 100

120 to 400

0.6 to 1.2

1000 to 3000

Selvaganapathy

130

1000

0.5 to 5

1900 to 19500

  • The herringbone structure is interesting but perhaps a bit outside the realm of most microfluidic oxygenators that do not rely on chaotic mixing structures to achieve highly efficient oxygen transfer.

We agree with the reviewer that “herringbone structure is perhaps a bit outside the realm of most microfluidic oxygenators that do not rely on chaotic mixing structures to achieve highly efficient oxygen transfer.”  This is related to the same reviewers’ point that our gaps are much wider.  In fact, this is some of the value of this manuscript, we believe.  We are not just investigating the range of gap heights and resulting shear that would be experienced in existing devices, but also anticipating that some designers will be tempted to use larger gaps and chaotic mixing in order to achieve efficiency at high flow rates.  This is a novel idea that has been proposed in other areas for improved transport and has been proposed for blood-contacting devices.  It is clear that larger gap devices with mixing elements can achieve effective transport, but the effects on hemolysis have not been studied.  This paper presents both early indications of the amount of hemolysis caused (negligible under these conditions) and suggests that this computational tool could help design devices that incorporate these elements.

  • One of the most significant concerns with the results presented in this manuscript is the lack of distinction between the various designs in terms of hemolysis rates (Figure 11.) The absence of a demonstrated ability to distinguish between different microfluidic geometries reduces confidence in the reported approach, as the positive control invoked by the authors is simply a length of tubing with high shear.  A more relevant and instructive positive control would be a microfluidic geometry that differs from the other designs by virtue of the presence of sudden expansions/contractions or sharp corners, rather than just a high shear tube.

We acknowledge the reviewer's concern regarding the lack of significant differences in hemolysis rates observed between the microfluidic designs (Figure 11).  While this may initially seem to limit the study's ability to differentiate between geometries, we believe the findings still hold value.

Table S3 summarizes the observed shear rates for each microfluidic device design. Importantly, despite these variations in shear rate, negligible hemolysis was consistently measured across all devices. Empirically determined hemolysis levels for all four microfluidic devices were remarkably low (< 2 ppm) and statistically indistinguishable from the negative control (p > 0.05) as shown in Figure 7. 

In this regard, the following text has been added to the manuscript: (line 529 to 540)

“This result indicates that even in high shear generated by devices such as Reduced Gap (~105,500 1/s) and Reduced Port (~150,800 1/s), significant hemolysis is not observed. This finding extends to a broader range of device geometries and operating conditions, suggesting that the risk of hemolysis induced by shear stress within microfluidic devices remains low even under harsher conditions.  This provides future researchers with greater design freedom when exploring novel microfluidic concepts, as long as anticipated shear rates fall within or below the ranges investigated in this study. The selection of a narrow tube as a positive control (~270,700 1/s) served a critical purpose.  By establishing a scenario with demonstrably high hemolysis levels, we validated our chosen assessment method.  This confirms that the employed techniques are capable of detecting very small levels of hemolysis, but just that the devices caused effectively negligible levels.”

  • Figure 9 appears to suggest that the hemolysis predicted for a given geometry is more dependent on the type of model used than on the dimensional properties and details of the design. This is a somewhat disconcerting conclusion; while the model optimization is important, it is ultimately of greater interest to explore various design approaches for microfluidic oxygenators and to enable their evaluation with respect to blood damage.

We completely agree.  The most valuable tool to the device designer is a single set of parameters that can be used.  Unfortunately, consensus on this has not yet occurred, even in the field of blood pumps, which has been applying similar hemolysis models for a longer time. The very foundation of these models – the experimental data used to derive them – varies considerably. The shear rate ranges employed and the methods used to apply shear stress differ significantly between various models. These discrepancies can potentially lead to weaker correlations between the models' predictions and the actual behavior of specific devices, particularly in the context of microfluidics studies.

It is true that the figure suggests a dependence of predicted hemolysis on the chosen model rather than solely on the microfluidic device design. While model optimization remains an ongoing effort, we acknowledge that the goal is to enable researchers to confidently explore and evaluate various microfluidic oxygenator designs with respect to potential blood damage.

In addressing this point, we would like to clarify the capabilities of the current model.  Our results indicate that while the model may not capture subtle differences in hemolysis between the microfluidic devices tested, it successfully differentiated the high hemolysis levels induced by the positive control. This suggests that the model can accurately predict the trend of hemolysis for microfluidic design iterations, provided the predicted hemolysis exceeds a certain threshold (IHPP).  Based on our data, this threshold is currently set at the level observed in the positive control.

As stated in the discussion, future work will focus on refining the model to improve its ability to predict hemolysis across a wider range of shear stress conditions and device geometries.  This will allow researchers to utilize the model with greater confidence in the design and evaluation of microfluidic oxygenators.

To convey the above-mentioned information, text has been added to the manuscript: (line 427 to 445)

“The PL model variants predicted hemolysis values across several orders of magnitude depending on which set of empirical parameters was used. This is discouraging as the ideal model would be one that accurately predicts the absolute value of hemolysis in any device. Unfortunately, consensus on this has not yet occurred, even in the field of blood pumps, which has been applying similar hemolysis models for a longer time. Significant variations exist in the shear rate ranges and stress application methods employed for model development. These discrepancies can lead to weakened correlations between predicted hemolysis and the actual behavior of specific devices, particularly in microfluidics. As observed, the predicted hemolysis can depend on the chosen model, not solely on the microfluidic design.

Despite these limitations, our study clarifies the current model's capabilities. While it might not capture subtle variations in hemolysis between the tested devices, it successfully identified the high hemolysis induced by the positive control. This suggests the model's ability to predict hemolysis trends for design iterations, provided the predicted hemolysis surpasses a certain threshold (IHPP) – currently defined based on the positive control data. As discussed further later on, future work will focus on refining the model for improved hemolysis prediction across a broader range of shear stress conditions and device geometries. This will enhance researchers' confidence in utilizing the model for microfluidic oxygenator design and evaluation.”

  • As stated in the conclusions, the syringe pump and test loop may be predominating the hemolysis in this analysis, calling into question the use of a syringe pump for such a study. Perhaps a more informative study could be conducted with a pump design closer to that used in ECMO circuits, and microfluidic oxygenator geometries that are more relevant for clinical-scale applications and oxygen transfer performance levels.

The reviewer raises a valid point regarding the potential influence of the syringe pump and test loop on hemolysis observations. We acknowledge that these components may contribute to the measured hemolysis levels.

To address this concern, we took several steps during the study design.  First, we ensured that the inner diameter of the syringe and loop tubing significantly exceeded the largest dimension of any microfluidic device. This approach theoretically minimizes the contribution of shear stress within the tubing compared to the microfluidic device itself. It is noteworthy that conventional ECMO circuits exhibit hemolysis around 10 ppm per pass, whereas the highest observed hemolysis in any of our microfluidic devices (including the test loop) was approximately 3 ppm per pass (Refer to manuscript: line 493 to 501).  This observation strengthens the claim that our experimental setup was appropriate for relevant levels of hemolysis detection.

We also explored the use of a peristaltic pump as an alternative.  However, this method introduced a confounding factor: the squeezing action of the rollers directly contributed to hemolysis, leading to significantly higher observed hemolysis levels compared to the syringe pump.  Additionally, peristaltic pumps offer lower resolution in flow control compared to syringe pumps, making it difficult to precisely regulate shear stress within the microfluidic devices.  Furthermore, the maximum shear stress in the peristaltic pump setup likely occurred within the pump itself, not necessarily within the microfluidic device under investigation.

Given these observations, we believe the syringe pump remained the most suitable choice for this study.  Our primary objective was to identify the IHPP for the test devices with an acceptable level of tolerance.  The controlled and precise flow rates achievable with a syringe pump facilitated this goal.

Round 2

Reviewer 2 Report

Comments and Suggestions for Authors

The revised manuscript “Empirical and Computational Evaluation of Hemolysis in a Microfluidic Extracorporeal Membrane Oxygenator (ECMO) Prototype” submitted to Micromachines by Imtiaz  et al. addresses many of the concerns raised in the initial review.  However, one of the main added elements, Table 1, is intended to address a major gap in the original manuscript, namely the need to cover higher flow and more clinically relevant microfluidic oxygenators.  However, the authors have chosen to include reports from each of three key groups that are not close to the highest flow rates they have reported.  Recent publications from each group report animal studies at much higher flow rates, and therefore not only the flow rates but the calculated shear stresses and other parameters would be much more relevant.  It is suggested that the authors consider updating the table to address one of the principal concerns of the original critique, namely that they should focus on the highest blood flow rates and most relevant (preclinical animal model) publications from each of the groups included in Table 1.  The method for calculating shear in this table should also be presented in detail.

Comments on the Quality of English Language

Minor editing required.
